# The Lignan-Rich Fraction from *Sambucus williamsii* Hance Exerts Bone Protective Effects via Altering Circulating Serotonin and Gut Microbiota in Rats

**DOI:** 10.3390/nu14224718

**Published:** 2022-11-08

**Authors:** Hui-Hui Xiao, Yu-Xin Zhu, Lu Lu, Li-Ping Zhou, Christina Chui-Wa Poon, Chi-On Chan, Li-Jing Wang, Sisi Cao, Wen-Xuan Yu, Ka-Ying Wong, Daniel Kam-Wah Mok, Man-Sau Wong

**Affiliations:** 1State Key Laboratory of Chinese Medicine and Molecular Pharmacology (Incubation), The Hong Kong Polytechnic University Shenzhen Research Institute, Shenzhen 518057, China; 2Research Centre for Chinese Medicine Innovation, The Hong Kong Polytechnic University, Hong Kong, China; 3Department of Applied Biology and Chemical Technology, The Hong Kong Polytechnic University, Hong Kong, China; 4School of Optometry, The Hong Kong Polytechnic University, Hong Kong, China; 5Centre for Eye and Vision Research (CEVR), 17W Hong Kong Science Park, Hong Kong, China

**Keywords:** osteoporosis, lignans, serotonin, TPH-1, gut microbiota

## Abstract

Our previous study revealed that the bone anabolic effects of the lignan-rich fraction (SWCA) from *Sambucus williamsii* Hance was involved in modulating the metabolism of tryptophan in vivo and inhibiting serotonin (5-HT) synthesis in vitro. This study aimed to determine how SWCA modulates bone metabolism via serotonin in vivo. The effects of SWCA were evaluated by using 4-month-old Sprague-Dawley (SD) ovariectomized rats. The serum levels of 5-HT and kynurenine, the protein expressions of tryptophan hydroxylase 1 (TPH-1) and TPH-2, the genes and proteins related to the 5-HT signaling pathway as well as gut microbiota composition were determined. SWCA treatment alleviated bone loss and decreased serum levels of serotonin, which was negatively related to bone mineral density (BMD) in rats. It suppressed the protein expression of TPH-1 in the colon, and reversed the gene and protein expressions of FOXO1 and ATF4 in the femur in OVX rats, while it did not affect the TPH-2 protein expression in the cortex. SWCA treatment escalated the relative abundance of *Antinobacteria* and modulated several genera relating to BMD. These findings verified that the bone protective effects of lignans were mediated by serotonin, and provided evidence that lignans might be a good source of TPH-1 inhibitors.

## 1. Introduction

Osteoporosis is a chronic and long-term skeletal metabolic disease and is twice as common in women as in men over 50 years of age. Osteoporosis fractures result in extensive morbidity, mortality, and increased health care cost with the progressive aging of a population [1]. Patients or potential patients with osteoporosis need long-term treatment or prevention. A number of skeletal anabolic and antiresorptive agents have become available to treat osteoporosis and related fractures; however, the uncertain and controversial adverse effects, namely uterus bleeding, breast cancer, thromboembolic events, atypical fractures, and osteonecrosis of the jaw, vis-à-vis the benefits of these therapies have become a substantial barrier to initiation of anti-osteoporosis therapy and to treatment adherence [2]. As estrogen deficiency is the major risk factor in the development of osteoporosis. Phytoestrogens, which are non-steroidal compounds from natural plants with estrogen-like biological activities, have attracted great interest in the field of bone and mineral metabolism. Phytoestrogens are naturally occurring compounds in a wide range of plant foods including soybean, oilseeds, cereals, legumes, fruits, vegetables, flaxseed, herbal medicines, and so on [3], and can be divided into several different groups, such as isoflavones, lignans, coumestans, and stilbenes. Phytoestrogens are a great source of potential anti-osteoporosis agents. 

The lignans are a class of phytoestrogens with anti-inflammatory, antioxidant, antitumor, anti-diabetes and anti-cardiovascular activities [4], while their effects on other chronic diseases, such as osteoporosis and breast cancer, remain controversial. The benefits of lignans on bone health through stimulating osteoblast proliferation or differentiation and suppressing osteoclastogenesis via modulating MAPK [5], NFκB [6], BMP2-SAMDs [7], or RANKL [8] signaling pathways, or activating estrogen receptor α/β [9], as well as improving bone mass and bone microarchitecture in ovariectomized (OVX) rat or mice models, have been reported by many preclinical experiments [5,10]. Although a cross-sectional study of postmenopausal women has revealed that the increased excretion of urine enterolactone, a bioactive enterolignan converted from flaxseed lignans by intestinal microbiota, is associated with high rate of bone loss [11], several clinical trials have not shown significant actions of flaxseed lignans on BMD or bone metabolic markers of postmenopausal women [12,13]. However, due to the limited number of clinical trials available, a conclusion is yet to be drawn. 

Our previous study revealed the bone beneficial effects of *Sambucus williamsii* Hance, and identified lignans as the major bioactive ingredients in *S. williamsii* [14,15]. The subsequent metabolomics study indicated that tryptophan metabolism was involved in mediating the bone protective effects of the lignan-rich fraction (SWCA) of *S. williamsii* [16]. SWCA treatment suppressed the synthesis of serotonin, one of the two major metabolites (kynurenine and serotonin) of tryptophan, by inhibiting protein expression of tryptophan hydroxylase 1 (TPH-1), the synthetic rate-limiting enzyme of serotonin in intestine, in RBL-2H3 cells (*tph-1* gene high expressing cells) [16]. This study aims to determine how SWCA modulates bone metabolism via serotonin in vivo. 

Serotonin (5-hydroxytraptamin, 5-HT) is best known as a neurotransmitter that mediates a wide range of central functions including behavior, mood, blood pressure, sleep, and thermoregulation. In the last two decades, multiple reports have indicated that serotonin plays an important role in bone metabolism. Tryptophan hydroxylase is the rate-limiting enzyme in 5-HT biosynthesis. More than 90% of the body’s serotonin is synthesized in the gastrointestinal tract through TPH-1 by enterochromaffin cells (ECs), and about 5% of the rest is synthesized in the brain through TPH-2 by clusters of neurons [17]. As serotonin cannot pass through the blood–brain barrier, the principal source of peripheral 5-HT is the gut. The gut-derived serotonin is transported into bone by platelets, and it reduces bone formation by inhibiting osteoblast proliferation via 5-HT1b receptors in pre-osteoblasts, while the brain-derived serotonin has positive effects on bone by decreasing the inhibitory action of sympathetic nerves on osteoblasts via acting on 5-HT2c receptors in ventromedial hypothalamic neurons [18]. In the intestine, ECs are exposed to enormous microbes and microbial products. Although gut microbiota does not appear to make 5-HT directly, it promotes colonic TPH-1 expression and 5-HT concentration in germ-free mice colonized with human gut microbiota through an effect of short-chain fatty acids (SCFAs) on ECs [19]. Thus, gut microbial composition is important to the enteric 5-HT production. 

Based on our previous studies, we hypothesized that SWCA could modulate 5-HT synthesis via gut TPH-1 and microbial composition to exert bone protective effects indirectly. In this study, the bone protective effects of SWCA were evaluated in OVX rats. The serum levels of serotonin and kynurenine, the protein expression of TPH-1 in the colon and TPH-2 in the cortex, the genes and proteins related to the 5-HT signaling pathway in the femur as well as the gut microbiota composition were determined to explore how SWCA exerted bone protective effects via serotonin in an estrogen-deficient animal model. 

## 2. Materials and Methods

### 2.1. Reagents and Materials

(7R,8S)-2,3-dihydro-2-(4–hydroxy-3-methoxyphenyl)-7-methoxy-5-(3-methoxy-1-(propenyl)-3-benzofuranomethanol (M1), Tetrahydrodehydrodiconiferyl alcohol (M2), 7-hydroxy-secoisolariciresinol (M3) and Secoisolariciresinol (M4) were prepared by Jiangsu Yongjiang Pharmaceutical Technology Co., LTD. (Taizhou, Jiangsu, China) with a purity >98%. Phosphate-buffered 10% formalin was ordered from Thermo Scientific (Waltham, MA, USA). Serotonin, L-kynurenine, and 5-methoxytryptamine (internal standard) were purchased from Sigma-Aladrich (St. Louis, MO, USA). D4-L-Kyn was purchased from TLC PharmaChem (ON, Canada). D4-Serotonin was obtained from Toronto Research Chemicals (Toronto, ON, Canada). HPLC grade acetonitrile, ethanol, and formic acid were purchased from Fisher Scientific (San Jose, CA, USA).

*S. williamsii* Hance was collected in Xinbin City, Liaoning Province, in the northeast of China in May 2017, and its identity was authenticated in accordance with Chinese Bencao (1999 Edition). A voucher specimen (SWSZ-2017) was deposited at the Shenzhen Institute of the Hong Kong Polytechnic University (Shenzhen, China). 

### 2.2. Preparation and Quality Control of SWCA

The dried ramulus of *S. williamsii* (150 kg) was chopped into small pieces and then hot refluxed with 60% aqueous ethanol (1200 L) twice, each time for 2 h. The extract was filtered and combined, and then concentrated to 150 L in a rotary evaporator under negative pressure. Finally, the extract was subjected to an HP-20 macroporous adsorptive resin column to give a lignan-rich fraction SWCA (50% aqueous ethanol eluate). SWCA is a purified fraction from SWC, and the origin of SWC was described in our previous study [14]. The yield of SWCA was 0.26% (*w*/*w*).

According to our previous study [20], the quality of SWCA was characterized by its four major components (M1 to M4) using an ACQUITY ultra-performance liquid chromatography (UPLC) H-Class system coupled to a Xevo TQD mass spectrometer (Waters Corp. Milford, CT, USA). The contents of the four markers in SWCA were determined to be 41.2, 27.2, 8.6 and 6.3 mg/g, respectively. The chromatographic conditions were consistent with those described in our previous study [21]. SWCA is a mixture of lignans, and more than thirty lignans were identified from it in our previous studies [15,20]. However, due to the limited amount of some of them and the bioactivities, M1 to M4 were selected and purified to be the markers.

### 2.3. Animals and Administration

Sprague-Dawley (SD) female rats (4-month old) were supplied by Beijing Vital River Laboratory Animal Technology Co., Ltd. (Beijing, China). The room temperature was maintained at 23 ± 2 °C with a humidity of 40–60%. An alternating 12 h light/dark cycle was set and the rats had free access to distilled water. The rats were pair-fed with a phytoestrogen-free diet (D00031602, Research Diets Inc., New Brunswick, NJ, USA) to eliminate interactions of diet and CA. The experimental procedures were approved by the Animal Ethics Committee of The Hong Kong Polytechnic University (No. 180402). 

After acclimatization for 5 days, the rats were randomly divided into Sham rats (10 rats) and OVX rats (38 rats). The Sham rats were sham operated, and the OVX rats were ovariectomized. After recovery for 10 days, the OVX rats were randomly divided into four groups and orally administrated with vehicle (ultrapure water, OVX, *n* = 10), 140 mg/kg SWCA extract (CAL, *n* = 10), 280 mg/kg SWCA extract (CAH, *n* = 10), or intramuscularly injected with 1.8 μg/kg Teriparatide (recombinant human parathyroid hormone analog, *n* = 8). The Sham rats were treated with vehicle. PTH was selected as the positive control in our study as it is a bone anabolic agent normally recommended for patients at high risk of fracture [22]. After 10 weeks of intervention and 12 h of fasting, all rats were anesthetized by intraperitoneal injection of chloral hydrate (5%, 0.7 mL/100 g) for blood collection, and then sacrificed for isolation of tissue samples, including the femur, tibia, uterus, colon, and cortex. 

### 2.4. Biochemical Analysis

Serum calcium (Ca), phosphorus (P), and alkaline phosphatase (ALP) levels were measured by commercially available kits (Shanghai Kehua Bio-Engineering Co., Ltd., Shanghai, China). The serum levels of bone metabolic markers of osteocalcin (OCN) and collagen type I C-telopeptide fragments (CTX-I) were determined according to the protocol of commercial enzyme linked immunosorbent assay (ELISA) kits (Signalway Antibody, Greenbelt, MD, USA).

### 2.5. Micro-Computed Tomography (Micro-CT) Analysis of Bone Properties

Bone properties of the distal femur and proximal tibia were determined by micro-CT (μCT40, Scanco Medical AG, Zurich, Switzerland). The trabecular bone was scanned at a resolution of 21 μm using an energy of 70 kVp and intensity of 110 μA, with an integration time of 300 ms. A total of 210 continuous slices were scanned from the metaphyseal growth plate to the metaphysis, and 100 slices were scanned starting from the same landmark (disappearing of the growth plate) for contouring the volume of interest (VOI) at the threshold value of 375 to evaluate the morphological properties. Bone mineral density (BMD, mg HA/ccm) and bone biological parameters, including trabecular bone number (Tb.N, mm^−1^), trabecular bone separation (Tb.Sp, mm), trabecular bone thickness (Tb.Th, mm), bone volume over total volume (BV/TV, %), connectivity density (Conn.D, mm^−3^), and structure model index (SMI), were calculated using a 3D direct model generated from VOI images. 

### 2.6. Bone Histology and Colon Immunohistology

The colon tissue was fixed by phosphate-buffered 10% formalin for 2 days. After embedding in paraffin by standard histological procedures, the samples were trimmed and sectioned in 3 μm thicknesses, stained with hematoxylin and eosin (H&E), and finally subjected to histological analysis by a light microscope equipped with a CCD camera (Nikon Co., Ltd., Tokyo, Japan) at 40× magnification. 

To detect TPH-1 protein in the colon, immunohistological analysis was performed on the formalin-fixed and paraffin-embedded tissue section using primary antibodies of anti-TPH-1 at a dilution of 1:200 (ImmunoWay Biotechnology, Newark, DE, USA). After incubation with biotinylated anti-rabbit secondary antibody (ZSGB-Bio, Beijing, China), the immunoreactivity was visualized by incubating the sections with a DAB chromogen kit, followed by counterstaining with hematoxylin. Finally, the sections were viewed with a Nikon Ni-U fluorescence microscope (Nikon Co., Ltd., Tokyo, Japan) and analyzed using Image-Pro Plus software (Media Cybernetics, Rockville, MD, USA).

### 2.7. Quantification of Plasma Serotonin and Kynurenine

To measure the levels of rat plasma serotonin (5-HT) and kynurenine (Kyn), the ultra-performance liquid chromatography tandem triple quadrupole mass spectrometry (UPLC-TQD-MS) method was established. D4-serotonin and D4-L-kynurenine (D4-L-Kyn) were used as “surrogate analytes” to demonstrate the linear relationship between the peak area of standard to IS. “Surrogate analytes” [23] were introduced to avoid the interference from endogenous 5-HT and Kyn on linearity, precision, accuracy, recovery, and matrix effects. The concentration ranges of the calibration curve for D4-serotonin and D4-L-Kyn were 1.95, 7.80, 31.20, 62.50, 125.00, 250.00, 500.00, 1000.00, 2500.00 ng/mL, and 15.60, 31.25, 62.50, 125.00, 250.00, 500.00, 1000.00, 2500.00, 5000.00 ng/mL, respectively. 

The plasma samples were removed from −80 °C storage and thawed at room temperature. A total of 100 μL of plasma was mixed with 10 μL internal standard (5-Methoxytryptamine, 1 μg/mL) and deproteinized by 300 μL methanol. After centrifugation at 14,000 rpm for 20 min, the upper layer of 200 μL was transferred to a clean Eppendorf and evaporated by a stream of nitrogen. The dried residue was reconstituted with 200 μL of H_2_O, and then 5 μL was injected into the UPLC-TQD-MS systems for analysis.

The analysis was performed using an ACQUITY UPLC H-CLASS system coupled to a Xevo TQD mass spectrometer (Waters Corp., Milford, CT, USA). LC separation was performed on an Acquity UPLC HSS T3 column (2.1 × 100 mm; 1.7 μm; Waters Corp., Milford, CT, USA). The mobile phase was maintained at a flow rate of 0.3 mL/min, containing 0.1% (*v*/*v*) formic acid water (A) and 0.1% (*v*/*v*) formic acid methanol (B). The elution gradients were as follows: 0–2.5 min, 10% B; 2.5–4.0 min, 10–70% B; 4.0–6.0 min, 70% B; 6.0–8.0 min, 70–10% B; 8.0–10 min, 10% B. The temperatures of the column and autosampler were controlled at 30 °C and 10 °C, respectively. The ESI source was set in positive ion mode (ESI^+^). Quantification was performed using multiple reaction monitoring (MRM) with *m/z* 177.4 → 160.1 transitions for serotonin, *m/z* 181.1 → 164.1 transitions for D4-serotonin, *m/z* 209.3 → 192.1 transitions for L-Kyn, *m/z* 213.0 → 196.0 transitions for D4-L-Kyn and m/z 191.0 → 174.0 transitions for the IS. The scan time per transition was 0.1 s. The optimal MS parameters were: capillary voltage of 3.0 kV, source temperature of 150 °C, and desolvation temperature of 350 °C. The curtain gas flow was 40 L/h, desolvation flow was 650 L/h, and con flow was 50 L/h. The collision voltages of serotonin, D4-serotonin, Kyn, D4-L-Kyn and IS were 6, 6, 9, 10 and 6 V, respectively. The data were processed using Masslynx V4.1 software (Waters Corp., Milford, CT, USA). 

### 2.8. Real-Time Quantitative Reverse Transcription-Polymerase Chain Reaction (RT-PCR)

The right femur of each rat was crushed under liquid nitrogen and total RNA was extracted according to the TRIzol manufacturer’s protocol (Invitrogen, Frederick, MD, USA). The cDNAs were transcribed with 2 μg of total RNA by using a High-Capacity cDNA Reverse Transcription Kit (Applied Biosystems, Waltham, MA, USA). The real-time PCR was performed using SYBR Green Supermix (Bio-Rad Laboratories Inc., Hercules, CA, USA) and an iCycler with an Iq5 Multicolor Real-Time PCR Detection System (Bio-Rad Laboratories Inc., Hercules, CA, USA). All primers used for performing RT-PCR are listed in Appendix A.

### 2.9. Western Blot Analysis

The femur was decalcified in 10% EDTA for 1 week. It was then washed twice with PBS and ground in a mortar with liquid nitrogen. Treated tissues were lysed with RIPA buffer (containing PMSF) for 45 min on ice. The pre-treated femur and cortex were lysed with RIPA buffer (containing PMSF) on ice for 90 min and 30 min, respectively. Lysates were centrifuged at 12,000 rpm for 20 min, and the protein concentration was measured by a BCA protein assay kit (Beyotime Biotechnology, Shanghai, China). Equal amounts of total protein were separated by 10% SDS-PAGE and transferred onto PVDF membranes. After blocking by 5% nonfat milk, the membranes were probed with primary antibody including ATF4, CREB, and FOXO1 (Abcam, Cambridge, UK), TPH1 and TPH2 (ImmunoWay, Biotechnology, Beijing, China), and actin (Beyotime Biotechnology, Shanghai, China) at 4 °C overnight, followed by rinsing 3 times with TBST. The PVDF membranes were than incubated with HRP-conjugated secondary antibody (1:2000, Abcam, Cambridge, UK), and the protein bands were visualized by enhanced chemiluminescence (ECL) reagent, and quantified by using ChemDoc XRS system (Bio-Rad Laboratories Inc., Hercules, CA, USA).

### 2.10. Hemostasis Assay

Blood was collected into a vacuum tube containing 0.109 mol/L sodium citrate (sodium citrate:blood = 1:9), and then centrifuged for 15 min at 3000 rpm. The supernatant was collected and stored at −20 °C for further analysis. The activated partial thromboplastin time (APTT), thromboplastin time (TT), prothrombin time (PT), and fibrinogen (FIB) were determined according to the procedure provided by the commercial kits (SUNBIO, Beijing, China) using a Coatron M1 Coagulation Analyzer (TECO Medical Instruments, Neufahrn, Germany).

### 2.11. 16S rRNA Gene Sequencing and Analysis

Fecal samples of OVX mice were collected after orally administrating CA extract for 10 weeks. Bacterial genomic DNA was extracted from 0.2 g rat fecal samples using the Stool Genomic DNA Extraction Kit (Solarbio Life Sciences, Beijing, China) according to the manufacturer’s instructions. The DNA quality was determined by 1% agarose gel electrophoresis. The 16S rRNA gene amplification and paired-end sequencing and data analysis were performed as previously described [21]. Briefly, the V3-V4 regions of bacteria 16S rRNA gene were amplified, purified, and sequenced using the Illumina MiSeq platform (Illumina, CA, USA) according to the standard protocols by Majorbio Bio-Pharm Technology Co., Ltd. (Shanghai, China), the sequences were analyzed using quantitative insights into microbial ecology (QIIME 1.9.1) software, and statistical analyzes were performed using R packages (V.2.15.3).

### 2.12. Statistical Analysis

The data were expressed as mean ± SEM. The data were checked for the assumptions of normality and equal variance, and then performed by GraphPad PRISM software package 8. To evaluate the actions of lignan-rich fractions in alleviating the effects of estrogen deficiency in ovariectomized rats, a one-way analysis of variance (ANOVA) was performed, followed by Tukey’s multiple comparisons test with a single pooled variance. The results with *p*-values of <0.05 were considered statistically significant. The correlation between serum serotonin and BMD was evaluated by linear regression, and the Pearson R^2^ was calculated. A value of *p* < 0.05 was considered to be statistically significant. 

The 16S rRNA sequence analysis was performed according to previous publications [24,25]. The diversity and richness of operational taxonomic unites (OTUs) were measured by Sobs index and Shannon index using the Student’s *t*-test. The principal coordinate analysis (PCoA) was calculated by Bray Curtis Distance analysis with an ANOSIM test. The analysis of significances between groups was performed by the Kruskal–Wallis H test followed with the Tukey–Kramer post hoc test. The clade of microbial community was analyzed by the LEfSe tool with LDA scores > 2 and shown as cladogram. The correlation of bone mineral density and microbial species was performed by a Spearman correlation heatmap. 

## 3. Results

### 3.1. SWCA Extract Improved Bone Properties without Altering Body Weight, Uterus Index or Serum Biochemistries in OVX Rats

As shown in Table 1, the body weight of OVX rats was significantly higher than that of sham rats, even when the two groups were pair-fed daily with the same amount of food (15 g/rat). After treatment for 10 weeks, neither low nor high doses of SWCA extract altered weight gain in OVX rats, which was similar to the treatment of PTH. 

As shown in Table 1, ovariectomy induced significant uterus atrophy in rats compared with those with the Sham operation, suggesting that the estrogen deficiency model was successfully established. Treatment of OVX rats with SWCA extract and PTH did not alter uterus weight gain. Moreover, ovariectomy and treatment with SWCA extract and PTH did not significantly alter the levels of serum calcium and phosphorus in rats. Ovariectomy induced a significant decrease in the serum osteocalcin (OCN) level and increase in serum C-terminal telopepitide fragments of type I collagen (CTX-I) in rats, while treatment with a low dose of SWCA extract significantly increased the serum OCN level and decreased CTX-I in OVX rats. The treatment with a high dose of SWCA and PTH also showed significant inhibiting effects on serum CTX-I. Although an increasing trend of the serum ALP level in OVX rats was observed when compared with Sham rats, the changes did not reach statistical significance. Treatment with SWCA extract and PTH slightly suppressed the high level of ALP in OVX rats, but the changes were not statistically significant. 

The trabecular BMD and bone microarchitectural parameters, including Tb.N, Tb.Sp, Tb.Th, BV/TV, Conn.D, and SMI of rats were significantly decreased in both the tibia and femur upon ovariectomy for 11 weeks (Table 1). A low dose of SWCA extract prevented the reduction in BMD at these two sites and improved bone microarchitecture, such as Tb.N, Tb.Sp and Coon.D, and SMI, in OVX rats, and these effects were similar to those of PTH on the trabecular bone. The 3D structures of distal femur and proximal tibia are visualized in Figure 1.

### 3.2. SWCA Extract Suppressed Serum Serotonin Level without Altering Kynurenine Level in OVX Rats

Serum serotonin (5-HT) in OVX group increased significantly (vs. Sham group), and treatment with SWCA extract and PTH significantly reduced serum 5HT levels in OVX rats (Figure 2A). In contrast, neither ovariectomy nor drug treatment altered the levels of serum kynurenine (Kyn) in rats (Figure 2B). In Figure 2C, the linear regression analysis showed that serum level of 5-HT was negatively correlated to BMD (R^2^ = 0.2024, *p* < 0.05) in rats.

### 3.3. SWCA Extract Modulated Gut-Derived Serotonin-Related Signaling Pathway in Femur

Gut-derived serotonin is crucial for bone remodeling. It can bind to the membrane 5HTR1b receptor of osteoblasts and then trigger the 5HTR1b/CREB/cyclins signaling cascade to influence osteoblast proliferation [26]. In the current study, the cascade signals related to gut-derived serotonin in bone, including 5HTR1b, ATF4, CREB, and FOXO1, were determined in the femur. As a low dose of SWCA exerted more significant effects on bone than the high dose of SWCA, CAL was selected for further signaling pathway study. Ovariectomy significantly increased the gene expression of 5HTR1b in rat bone, but the expression of 5HTR1b was not altered by drug treatment in OVX rats (Figure 3A). The gene and protein expressions of ATF4 and FOXO1 were significantly upregulated and downregulated in OVX rats, respectively, while CAL treatment significantly restored them to normal levels (Figure 3A,B). In contrast to the effects of CAL treatment, PTH treatment did not alter the gene expression of ATF4 or protein expression of FOXO1, but decreased the protein expression of ATF4 and increased the gene expression of FOXO1. OVX operation and treatment did not influence the gene and protein expressions of CREB. 

### 3.4. SWCA Extract Inhibited TPH-1 Protein Expression in the Colon, but Did Not Influence TPH-1 and TPH-2 Protein Expressions in the Brain Cortex

The immunohistological studies of TPH-1 protein expression in the colon, and TPH-1 and TPH-2 expressions in the brain cortex are presented in Figure 4. H&E staining showed no obvious differences in the histological features of the colon in all experimental groups. Immunohistological analysis of the colon showed that ovariectomy significantly increased the protein expression of TPH-1 in rats (vs. Sham group), while treatment of OVX rats with either CAL or PTH significantly suppressed the expression of TPH-1 in the colon. In contrast, ovariectomy did not alter the TPH-1 expression, but significantly decreased TPH-2 expression of the brain cortex in rats. The protein expressions of TPH-1 and TPH-2 in the brain cortex in rats were not altered in any of the drug treatment groups.

### 3.5. SWCA Extract Did Not Influence Coagulation

Serotonin plays a critical role in primary hemostasis and it accelerates the conversion of fibrinogen to fibrin [27]. In order to rule out the possibility that SWCA extract could exert adverse effects on coagulation, the activated partial thromboplastin time (APTT), thromboplastin time (TT), prothrombin time (PT), and fibrinogen (FIB) were measured. As shown in Figure 5, neither the coagulation times nor markers were affected by SWCA treatment, which indicated that SWCA exerted bone anabolic effects without deleterious consequences to hemostasis. 

### 3.6. SWCA Extract Slightly Altered the Microbial Composition

Since gut microbes are able to modulate the synthesis of mucosal 5-HT, the microbial composition was determined. As PTH was intramuscularly injected without direct interaction with gut microbiota, the microbiota was not studied in PTH treatment. The fecal samples of rats at baseline (0 week) were collected upon ovariectomy for 1 week and before the treatment. As shown in Figure 6, the microbial diversity slightly decreased in rats upon recovery from ovariectomy for 1 week; however, no significant differences in microbial diversity were found among the Sham, OVX and CAL groups. The microbial diversities of all groups increased significantly after 10 weeks of treatment compared with the OVX group at 0 week, yet no statistical differences in microbial diversity were found among the Sham, OVX and CAL groups after treatment for 10 weeks. The PCoA and PLS-DA analysis showed that the operational taxonomic units (OTUs) at week 10 were well separated from that at 0 week, suggesting the microbial composition had altered over time. At week 10, the bacteria in the CAL treatment group induced a significant separation from that of the Sham and OVX groups, indicating that CAL treatment had significant influence on intestinal bacterial composition.

### 3.7. SWCA Extract Significantly Altered Antinobacteria Phylum and Its Belonged Genus

CAL treatment for 10 weeks extremely enriched the abundance of *Antinobacteria* phylum in rat fecal samples (Figure 7A). In order to identify the clade of *Antinobacteria* phylum, the microbial community from phylum to genus level was analyzed by the LEfSe tool with an LDA score > 2. As shown in Figure 7B, the LEfSe analysis showed that a series of microbial taxa had contributed to the enrichment of *Antinobacteria* phylum, including class: *Antinobacteria* → order: *Coriobacteriales* → family: *Coriobacteriaceae* → genus: *Adlercreutzia*, *norank_f_Coriobacteriaceae*, *Parvibacter*, *Enterorhabdus* and *Collinsella* (Figure 7B). The five genera that contributed to the enrichment of *Antinobacteria* phylum abundance are shown in Figure 7C. All these genera were significantly reduced in the OVX groups when compared with the Sham groups, and were highly induced in OVX rats treated with CAL (Figure 7C).

### 3.8. SWCA Extract Modulated the Microbial Genera Correlated with BMD

In order to identify the gut microbial genera that were correlated to BMD, Spearman’s correlation analysis was performed. As shown in Figure 8, BMD of rats was negatively correlated with *[Eubacterium]_coprostanoligenes_group*, *Ruminococcaceae_UGC-014*, *Butyrivibrio*, *Lachnospiraceae_NK4A136_group* and *Clostridium_sensu_stricto_1*, and positively correlated with *Alistipes*. After treatment for 10 weeks, CAL significantly decreased the abundance of *Ruminococcaceae_UGC-014* and *[Eubacterium]_coprostanoligenes_group*, which might account for the bone protective effects of SWCA.

## 4. Discussion

Our previous metabolomics study revealed that the lignan-rich fraction of *S. williamsii* Hance could restore the tryptophan level of ovariectomized rats and inhibit gene and protein expressions of TPH-1 as well as suppress the serotonin synthesis in RBL-2H3 cells, thereby accounting for the bone protective effects of the fraction. In the present study, we confirmed that lignan-rich fraction SWCA treatment suppressed the TPH-1 protein expression in the colon, but not TPH-2 protein expression in the brain cortex, and led to the low level of serum serotonin, which was correlated with the bone protective actions of SWCA in rats. SWCA treatment reversed the gene and protein expressions of FOXO1 and ATF4, two key players in the serotonin receptor 5HTR1b/CREB/cyclins signaling cascade in the femur of OVX rats. Apart from its action on TPH-1 in the colon, SWCA also altered the gut microbial composition, and highly induced the relative abundance of *Antinobacteria* in the gut microbiota of OVX rats. In addition, several genera, such as *[Eubacterium]_coprostanoligenes_group*, *Ruminococcaceae_UGC-014*, *Butyrivibrio*, *Lachnospiraceae_NK4A136_group*, *Clostridium_sensu_stricto_1*, and *Alistipes*, were found to be correlated with the BMD of female rats. Furthermore, SWCA treatment did not affect the coagulation function. Our results indicated that the target of SWCA for achieving bone protective effects was in the intestine, such as TPH-1 in the colon for synthesis of gut-derived serotonin as well as gut microbiota composition.

An increasing number of cohort studies suggests that use of selective serotonin reuptake inhibitors (SSRIs), which are first-line antidepressants, is associated with increased extracellular 5-HT level and decreased BMD as well as increased fracture risk [28]. In a recent cohort study of 388,979 individuals, Kang et al. confirmed again that the use of serotonin reuptake inhibitors was associated with an increased risk of bone loss in both men and women with an average age of 58 [29]. However, a cross-sectional study of 45 women (mean age: 37.64) suffering from major depressive disorder did not show significant differences in the mean BMD in lumber spine, left hip, and right hip compared with healthy controls (mean age: 38.1) [30]. The controversial results might be due to the differences in methodology, the age of subjects and the analysis method. In any case, concerns about bone health have been raised when administering SSRIs. As blocking gut-derived 5-HT biosynthesis might be benefit for bone health, Gerard et al. used LP533401, a small molecular inhibitor of TPH-1 that had been tested in clinical trials for irritable bowel syndrome, to treat osteoporosis in ovariectomized rodents, and proved that inhibiting gut-derived 5-HT biosynthesis could be a new anabolic treatment for osteoporosis [31]. In this study, we confirmed the bone protective effects of SWCA, and demonstrated that serum serotonin was negatively correlated to BMD in female rats. Our results were in agreement with those reported in a clinical study that involved 275 age-stratified women, which included 90 premenopausal women and 125 postmenopausal women who did not receive hormone therapy [32]. Our results suggested that significant reduction of serum levels of serotonin after SWCA treatment might account for the bone protective effects of SWCA in OVX rats. 

The gut-derived 5-HT is transported by platelets to bone, and affects osteoblast proliferation by acting as a cAMP/PKA/CREB signaling cascade via binding to 5HTR1b receptor, a G protein-coupled receptor, in pre-osteoblasts [33]. Under normal physiological conditions, FOXO1 is associated with both CREB and ATF4 in osteoblasts. However, elevated circulating serotonin inhibits the interaction of FOXO1 with CREB, while increasing its interaction with ATF4, which in turn leads to inhibition of bone formation [18]. In this study, SWCA treatment significantly suppressed TPH-1 protein expression in the colon in OVX rats, which accounted for the low serum level of 5-HT in the SWCA-treated rats. CAL treatment significantly downregulated the AFT4 gene and protein expressions in the femur, but upregulated the FOXO1 gene and protein expressions in OVX rats, which might decrease the interaction of FOXO1 with AFT4 while increasing the interaction with CREB to promote bone formation. In addition, CAL treatment did not influence the TPH-2 in the cortex. The above results further supported the fact that the bone protective effects of CAL treatment in OVX rats were mediated by suppressing circulation of 5-HT levels via inhibiting intestine TPH-1. 

Several studies have indicated that gut microbiota may regulate 5-HT synthesis and metabolism by affecting microbial metabolites such as SCFAs, cholate, and p-aminobenzoate [17,19,34]. In addition, gut microbiota may influence the development and progression of osteoporosis by regulating its own metabolites, host metabolism, drug metabolism, and intestinal barrier function [35]. Thus, the composition of gut microbiota was determined in this study. A few recent studies reported the modulating actions of lignans on gut microbial composition. Bader et al. [36] revealed that flaxseed lignan secoisolariciresinol diglucoside altered 3 bacterial phyla and 22 genera after 10-day treatment in eight-week-old female C57BL/6 mice. Taibi et al. [37] found that lignan-rich oilseeds could strongly influence the microbiota composition of younger and premenopausal females, and led to different enterolignan production in an *in vitro* fermentation experiment. Our results showed that the microbial composition changed over time. At the end point, CAL treatment extremely increased the abundance of *Antinobacteria* phylum and its belonged genera of *Adlercreuzia*, *Collinsella*, *norank_f_Coriobacteriaceae*, *Enterorhabdus*, and *Parvibacter*. *Adlercreutzia* [38] is known as an equol-producing genus isolated from human feces and it is capable of metabolizing daidzein to equol, a nonsteroidal estrogen with superior antioxidant activity and estrogenic activity [39]. Several studies revealed that *Collinsella* could produce butyric acid, acetic acid, formic acid, and lactic acid [40], modify host bile acids and cholesterol levels, and correlate to low dietary fiber intake and circulating insulin [40]. The involvement of the *Coriobacteriaceae* family in polyphenol metabolism and consumption, as well as its correlation with cholesterol absorption, synthesis, and excretion were reported [41]. Sun et al. reported that *Enterorhabdus* and *Parvibacter* were positively correlated with the amount of intestinal butyric acid and valeric acid in tea polyphenols treated ethanol-induced alcoholic liver injury in mice [42]. Our results and the above publications indicated that the microbe modulated by SWCA treatment might be involved in lignan transfer, and bile acid and cholesterol metabolism.

In addition, some bacteria, including the *Lachnospiraceae_NK4A136_group*, *Butyrivibrio Ruminococcaceae_UGC-014*, *[Eubacterium]_coprostanoligenes_group*, *Alistipes*, and *Clostridium_sensu_stricto_1*, were found to be correlated with BMD in this study. Dan et al. demonstrated that Tiansi Liquid, a traditional Chinese herbal medicine used to treat depression, increased the relative abundance of a *Lachnospiraceae_NK4A136_group* and the serum 5-HT levels in a hydrocortisone-induced depression rat model [43], which suggested that the *Lachnospiraceae_NK4A136_group* was positively correlated to 5-HT. Our present study confirmed that the increased abundance of the *Lachnospiraceae_NK4A136_group* was coupled with the increased level of serum serotonin in OVX rats, which were both negatively related to BMD. Tu et al. [44] reported that the abundance of *Ruminococcaceae_UCG_014* was lower in Sham mice while it was enriched in OVX mice. Similarly, our results showed that OVX induced high abundance of *Ruminococcaceae_UCG_014*, while CAL treatment restored the changes to the Sham level. Rothhammer et al. [45] revealed that the gut microbiota generating metabolites of tryptophan was involved in gut microbiome-derived neurodevelopment diseases, and Chen et al. [46] showed that tryptophan was negatively linked to *[Eubacterium]_coprostanoligenes_group* in rat feces with kidney yang deficiency syndrome. Moreover, our study demonstrated that the *[Eubacterium]_coprostanoligenes_group* was negatively correlated to BMD, while BMD was negatively correlated to serum 5-HT. These studies and our results suggested that the *[Eubacterium]_coprostanoligenes_group* might have a positive correlation with gut-derived serum serotonin. All the above findings indicated that the genera of the *Lachnospiraceae_NK4A136_group* and *[Eubacterium]_coprostanoligenes_group* might indirectly affect bone metabolism via modifying the synthesis of gut-derived 5-HT. 

## 5. Conclusions

Our present rat study indicated that the lignan-rich fraction from *S. williamsii* might exert bone protective effects indirectly by suppressing serotonin synthesis via inhibiting the intestinal TPH-1 protein and modulating gut microbiota composition. The limitation of the present study was that there was no direct evidence for the modulating effects of lignan-altered microbiome on bone metabolism and 5-HT synthesis. Further studies of SWCA treated-feces transplantation to antibiotic pre-intervened mice are needed to determine the actions of lignan-induced microbiome on bone. In addition, *in vitro* studies of major lignans from SWCA on 5-HT synthesis and affinity to TPH-1 are necessary to verify the direct actions of lignans on 5-HT. 

## Figures and Tables

**Figure 1 nutrients-14-04718-f001:**
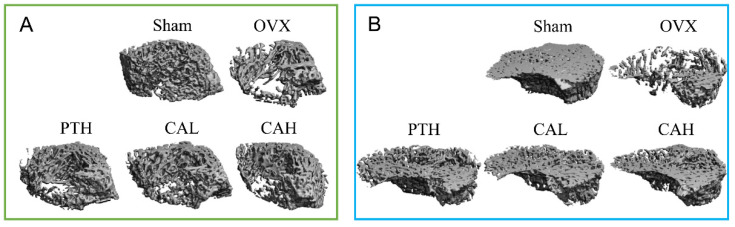
3D visualized structure of distal femur and proximal tibia. (**A**) Distal femur; (**B**) proximal tibia. The rats were orally administrated with vehicle (Sham, OVX), 140 mg/kg SWCA extract (CAL), 280 mg/kg SWCA extract (CAH), or intramuscularly injected with 1.8 μg/kg Teriparatide (PTH) for 10 weeks.

**Figure 2 nutrients-14-04718-f002:**
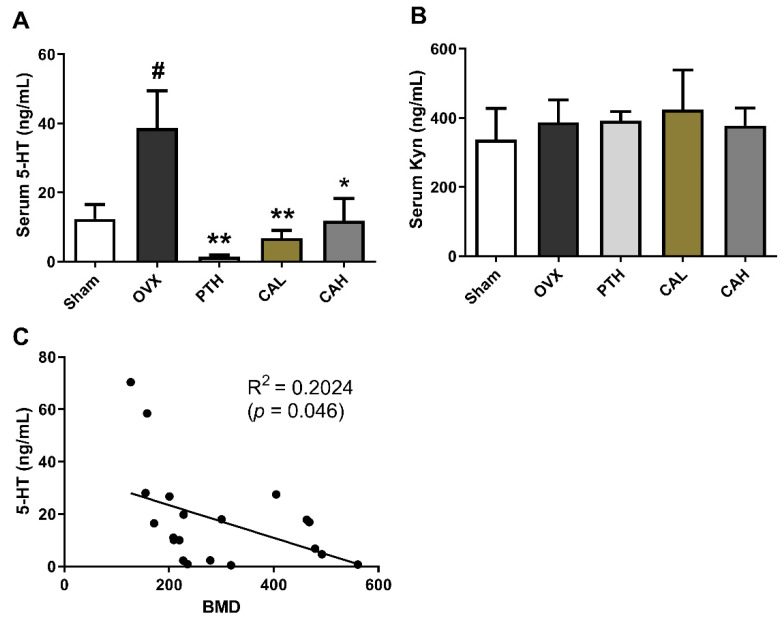
The effects of SWCA extract on serum levels of serotonin (5-HT) and kynurenine (Kyn) in ovariectomized rats treated with PTH and SWCA extract, and the correlation between 5-HT and BMD. (**A**) Serum 5-HT level; (**B**) serum Kyn level; (**C**) the correlation between 5-HT and bone mineral density (BMD). The rats were orally administrated with vehicle (Sham, OVX), 140 mg/kg SWCA extract (CAL), 280 mg/kg SWCA extract (CAH) or intramuscular injected with 1.8 μg/kg Teriparatide (PTH) for 10 weeks. Data are presented as the mean ± SEM, *n* = 6–8. # *p* < 0.05 vs. Sham, * *p* < 0.05, ** *p* < 0.01 vs. OVX by one-way ANOVA followed by Tukey’s post-hoc test. Pearson R^2^ was calculated by linear regression analysis. *p* < 0.05 was considered to be statistically significant.

**Figure 3 nutrients-14-04718-f003:**
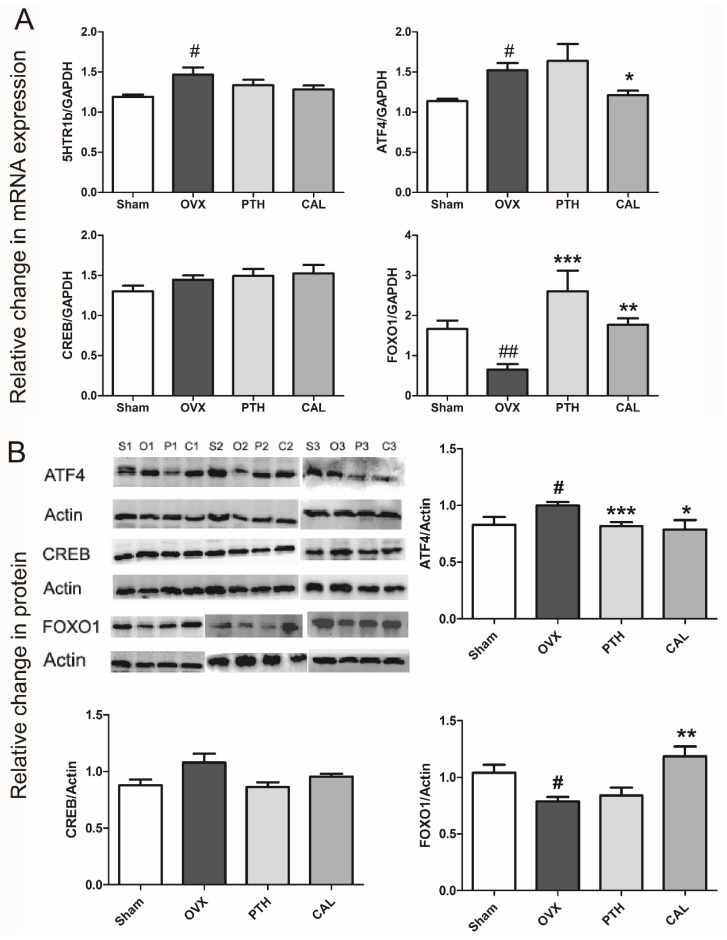
The effects of SWCA extract on gene and protein expressions of gut-derived serotonin-related signaling pathway in femur. (**A**) Gene expression; (**B**) protein expression. The rats were orally administrated with vehicle (Sham, OVX), 140 mg/kg SWCA extract (CAL), 280 mg/kg SWCA extract (CAH) or intramuscularly injected with 1.8 μg/kg Teriparatide (PTH) for 10 weeks. Data are presented as the mean ± SEM, *n* = 8–10. # *p* < 0.05, ## *p* < 0.01 vs. Sham, * *p* < 0.05, ** *p* < 0.01, *** *p* < 0.001 vs. OVX by one-way ANOVA followed by Tukey’s post-hoc test.

**Figure 4 nutrients-14-04718-f004:**
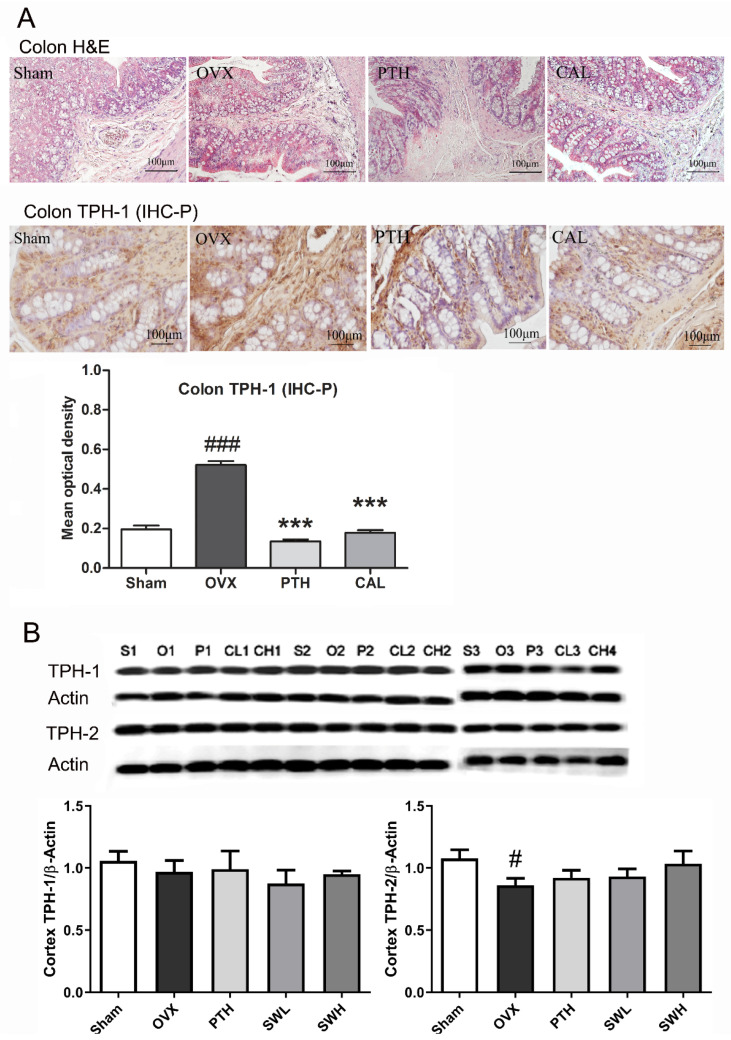
The effects of SWCA extract on colon and brain cortex TPH expressions. (**A**) The H&E staining and immunohistological staining of TPH-1 in colon sections at 100× magnifications; (**B**) the protein expressions of TPH-1 and TPH-2 in brain cortex. The rats were orally administrated with vehicle (Sham, OVX), 140 mg/kg SWCA extract (CAL), 280 mg/kg SWCA extract (CAH) or intramuscularly injected with 1.8 μg/kg Teriparatide (PTH) for 10 weeks. Data are presented as the mean ± SEM, *n* = 6–8. # *p* < 0.05, ### *p* < 0.001 vs. Sham, *** *p* < 0.001 vs. OVX by one-way ANOVA followed by Tukey’s post-hoc test.

**Figure 5 nutrients-14-04718-f005:**
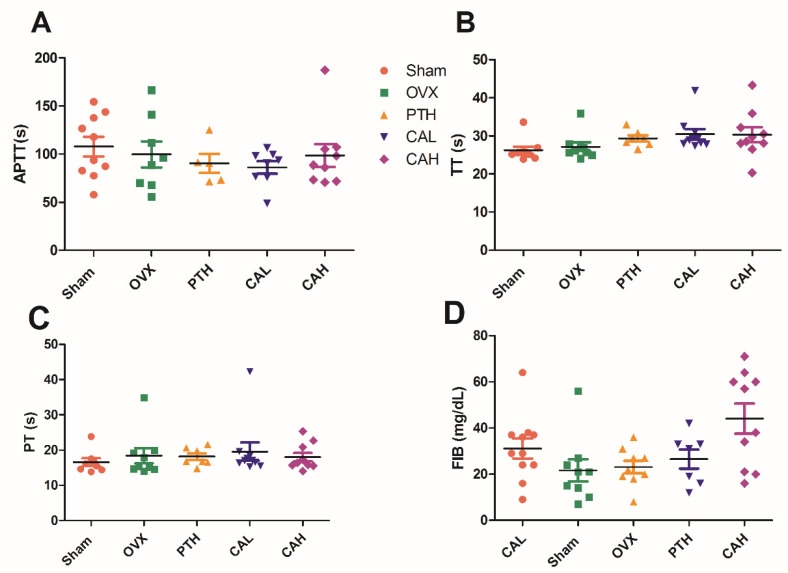
The effects of SWCA extract on coagulation. (**A**) APTT, activated partial thromboplastin time; (**B**) TT, thromboplastin time, (**C**) PT, prothrombin time; (**D**) FIB, fibrinogen. The rats were orally administrated with vehicle (Sham, OVX), 140 mg/kg SWCA extract (CAL), 280 mg/kg SWCA extract (CAH) or intramuscularly injected with 1.8 μg/kg Teriparatide (PTH) for 10 weeks. Data are presented as the mean ± SEM, *n* = 6–10. Group comparisons were performed with one-way ANOVA followed by Tukey’s post-hoc test. APTT, partial thromboplastin time; TT, thromboplastin time; PT, prothrombin time; FIB, fibrinogen.

**Figure 6 nutrients-14-04718-f006:**
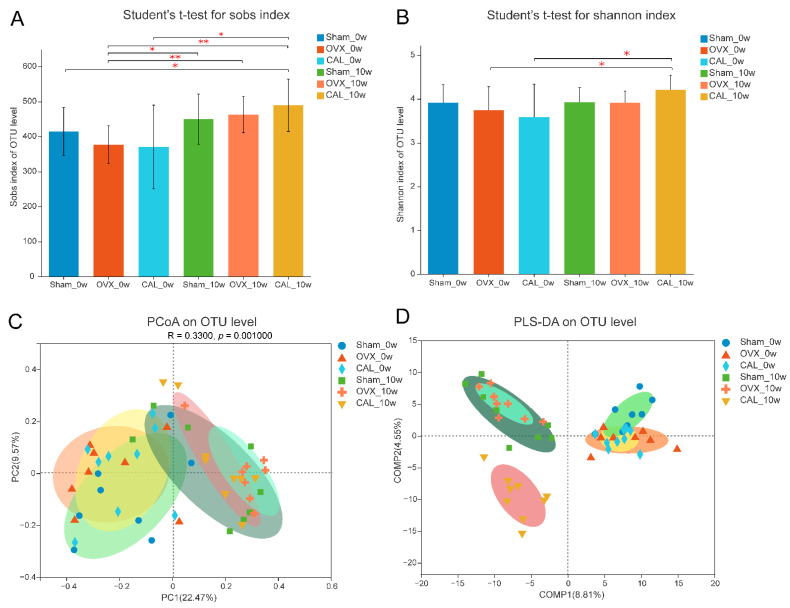
Sobs and Shannon index, and PCoA and PLS-DA score plots based on operational taxonomic unit (OTU) abundance. (**A**) Sobs index; (**B**) Shannon index; (**C**) Principal coordinate analysis (PCoA); (**D**) Partial least squares-discriminant analysis (PLS-DA). The rats were orally administrated with vehicle (Sham, OVX) or 140 mg/kg SWCA extract (CAL) for 10 weeks. Statistical analysis was performed by Student’s *t*-test (*n* = 8–9) in (**A**,**B**), and the significant difference between two groups is presented as * *p* < 0.05 and ** *p* < 0.01.

**Figure 7 nutrients-14-04718-f007:**
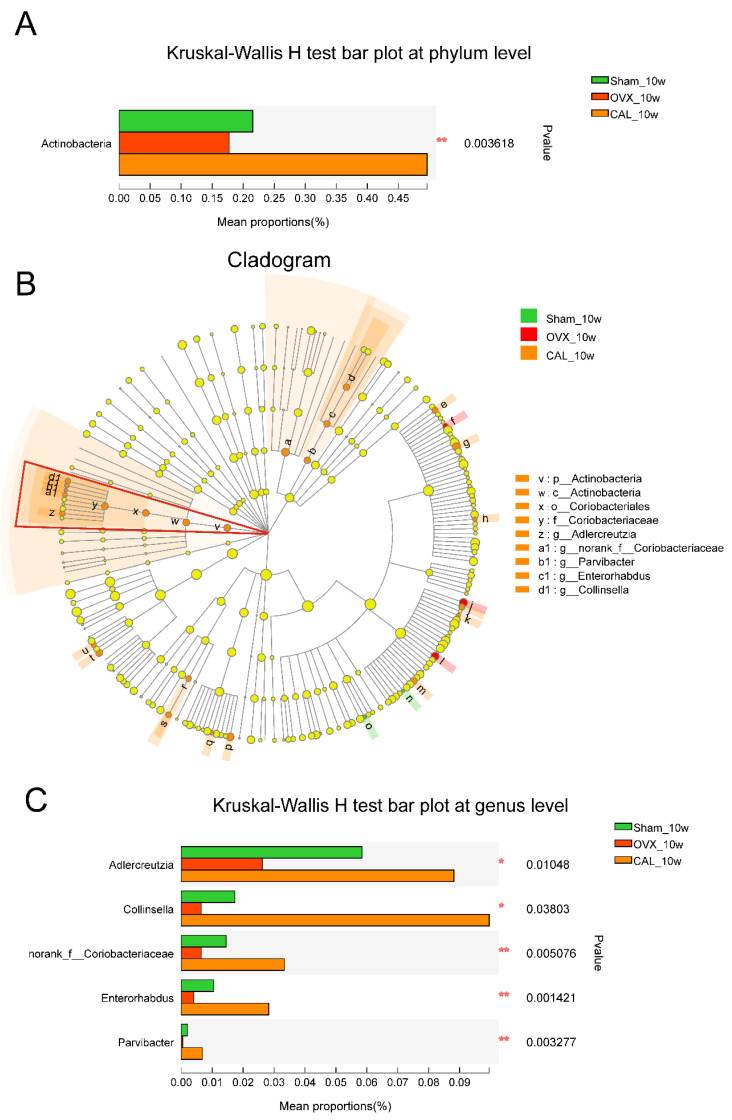
The significantly altered bacterial taxa at phylum and genus levels and the LEfSe (Linear discriminant analysis effects size) analysis of gut microbiota after treatment for 10 weeks. (**A**) Significantly altered bacteria at phylum level; (**B**) cladogram of LEfSe analysis from phylum to genus with LDA score > 2, nodes of the same color indicate microbial taxa that are significantly enriched in the corresponding group and has a significant impact on the difference between groups, while the light yellow nodes indicate microbial taxa that do not differ significantly in different groups or have no significant effect on differences between groups; (**C**) significantly altered genera in *Antinobacteria* phylum. The rats were orally administrated with vehicle (Sham, OVX) or 140 mg/kg SWCA extract (CAL) for 10 weeks. Statistical analysis was performed by the Kruskal-Wallis H test *(n* = 8–9) followed by the Tukey-Kramer test, and significant difference is presented as * *p* < 0.05, ** *p* < 0.01.

**Figure 8 nutrients-14-04718-f008:**
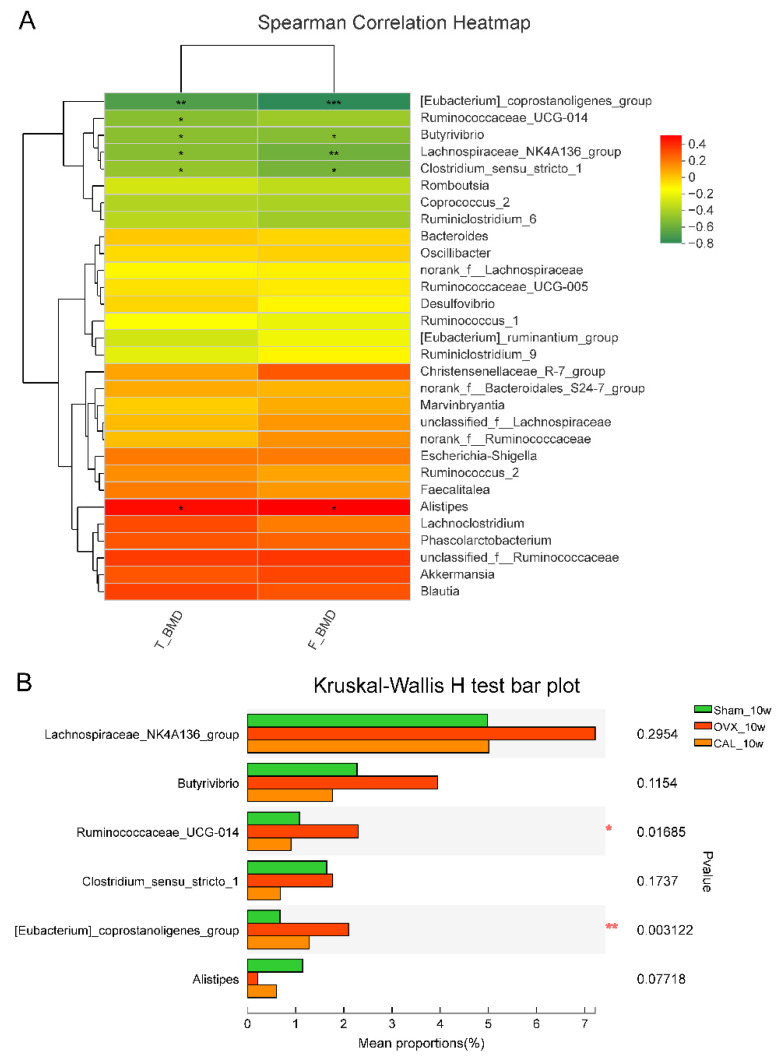
The correlation between relative abundance of top 30 genera and BMD of both tibia and femur, and the bar chart of the genera significantly related to BMD. (**A**) Spearman correlation heatmap. Red, positive correlation; Green, negative correlation. (**B**) Bar chart of genera related to BMD. The rats were orally administrated with vehicle (Sham, OVX) or 140 mg/kg SWCA extract (CAL) for 10 weeks. Statistical analysis was performed by the Kruskal-Wallis H test (*n* = 8–9) followed by the Tukey-Kramer test, and significant difference is presented as * *p* < 0.05, ** *p* < 0.01 and *** *p* < 0.001.

**Table 1 nutrients-14-04718-t001:** Effects of SWCA extract on bone properties, body weight, uterus index, and serum biochemistries.

Parameters		Sham	OVX	PTH	CAL	CAH
Tibia	BMD (mgHA/cm^3^)	468.1 ± 15.4 ^a^	167.4 ± 7.4 ^c^	245.1 ± 15.9 ^b^	209.0 ± 5.3 ^bc^	183.8 ± 10.5 ^c^
	Tb.N (mm^−1^)	5.50 ± 0.13 ^a^	1.55 ± 0.10 ^c^	2.35 ± 0.16 ^b^	2.10 ± 0.13 ^bc^	1.76 ± 0.14 ^c^
	Tb.Sp (mm)	0.141 ± 0.006 ^c^	0.653 ± 0.042 ^a^	0.416 ± 0.028 ^b^	0.497 ± 0.034 ^ab^	0.568 ± 0.045 ^a^
	Tb.Th (mm)	0.121 ± 0.002 ^a^	0.084 ± 0.001 ^c^	0.090 ± 0.002 ^b^	0.085 ± 0.001 ^bc^	0.085 ± 0.001 ^bc^
	BV/TV (%)	0.575 ± 0.017 ^a^	0.117 ± 0.008 ^c^	0.212 ± 0.021 ^b^	0.166 ± 0.007 ^bc^	0.128 ± 0.012 ^c^
	Conn. D (mm^−3^)	55.9 ± 2.4 ^a^	17.4 ± 2.1 ^c^	29.7 ± 3.1 ^b^	26.4 ± 1.4 ^bc^	19.6 ± 2.3 ^c^
	SMI	−3.41 ± 0.48 ^c^	1.74 ± 0.1 ^a^	0.89 ± 0.24 ^b^	1.36 ± 0.16 ^ab^	1.88 ± 0.10 ^a^
Femur	BMD (mgHA/cm^3^)	475.2 ± 18.4 ^a^	189.5 ± 11.3 ^c^	265.2 ± 16.6 ^b^	233.1 ± 5.6 ^bc^	207.6 ± 9.6 ^c^
	Tb.N (mm^−1^)	5.17 ± 0.19 ^a^	1.68 ± 0.08 ^c^	2.21 ± 0.27 ^b^	2.02 ± 0.12 ^bc^	1.69 ± 0.06 ^bc^
	Tb.Sp (mm)	0.179 ± 0.014 ^c^	0.685 ± 0.032 ^a^	0.516 ± 0.065 ^b^	0.557 ± 0.035 ^b^	0.697 ± 0.017 ^a^
	Tb.Th (mm)	0.116 ± 0.006 ^a^	0.086 ± 0.001 ^b^	0.089 ± 0.002 ^b^	0.088 ± 0.002 ^b^	0.088 ± 0.002 ^b^
	BV/TV (%)	0.528 ± 0.015 ^a^	0.169 ± 0.018 ^c^	0.251 ± 0.022 ^b^	0.222 ± 0.005 ^bc^	0.198 ± 0.014 ^bc^
	Conn. D (mm^−3^)	53.2 ± 3.0 ^a^	26.8 ± 2.3 ^c^	36.1 ± 2.9 ^b^	34.1 ± 0.8 ^bc^	29.0 ± 2.1 ^bc^
	SMI	−2.88 ± 0.25 ^b^	0.30 ± 0.22 ^a^	−0.37 ± 0.27 ^a^	0.04 ± 0.14 ^a^	0.46 ± 0.20 ^a^
Weight change (%)		19.9 ± 4.6 ^b^	37.7 ± 3.1 ^a^	48.0 ± 5.5 ^a^	37.1 ± 3.9 ^a^	35.2 ± 2.6 ^a^
Uterus index (UI, mg/g)		1.70 ± 0.12 ^a^	0.27 ± 0.01 ^b^	0.26 ± 0.02 ^b^	0.28 ± 0.03 ^b^	0.27 ± 0.02 ^b^
Serum Ca (mmol/L)		2.39 ± 0.03	2.33 ± 0.04	2.36 ± 0.02	2.33 ± 0.06	2.40 ± 0.04
Serum P (mmol/L)		2.13 ± 0.19	2.19 ± 0.10	2.04 ± 0.08	2.31 ± 0.07	2.02 ± 0.08
Serum ALP (ng/mL)		56.6 ± 7.6	77.7 ± 12.8	61.6 ± 6.8	65.7 ± 3.9	63.6 ± 5.3
Serum OCN (pg/mL)		362.8 ± 55.9 ^a^	213.2 ± 35.8 ^b^	272.1 ± 76.9 ^ab^	331.1 ± 32.1 ^a^	281.8 ± 28.1 ^ab^
Serum CTX-I (ng/mL)		0.49 ± 0.04 ^b^	0.67 ± 0.06 ^a^	0.47 ± 0.05 ^b^	0.49 ± 0.03 ^b^	0.41 ± 0.05 ^b^

Four-month-old SD rats were subjected to the following treatment for 10 weeks after ovariectomy: Sham, Sham-operated, vehicle-treated, *n* = 10; OVX, ovariectomized, vehicle-treated, *n* = 10; PTH, ovariectomized, intramuscular injected 1.8 μg/kg Teriparatide, *n* = 8; CAL, ovariectomized, low-dose SWCA fraction (140 mg/kg), *n* = 10; CAH, ovariectomized, high-dose SWCA fraction (280 mg/kg), *n* = 10. Data are presented as the mean ± SEM. With a give row, groups that do not share the same superscript letter are significantly different from each other with *p*-value < 0.05. BMD, bone mineral density; Tb.N, trabecular bone number; Tb.Sp, trabecular bone separation; Tb.Th, trabecular bone thickness; BV/TV, bone volume over total volume; Conn.D, connectivity density; SMI, structure model index; Ca, calcium; P, phosphorus; ALP, alkaline phosphatase; OCN, osteocalcin; CTX-I, C-terminal telopepitide fragments of type I collagen.

## Data Availability

The raw sequencing data are available at https://www.ncbi.nlm.nih.gov/sra/PRJNA693397 after the indicated release date.

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
