# Peer review of "The Lignan-Rich Fraction from Sambucus williamsii Hance Exerts Bone Protective Effects via Altering Circulating Serotonin and Gut Microbiota in Rats"

_nutrients, 2022, doi:10.3390/nu14224718_

Round 1

Reviewer 1 Report

A brief summary:
This study aims to determine how the lignan rich fraction (SWCA) from Sambucus williamsii Hance modulates bone metabolism via serotonin in vivo. The findings of Authors supported that the bone protective effects of lignans were mediated by serotonin, and provided evidence that lignans might be a good source of TPH-1 inhibitors.

General concept comments:

1. The manuscript is clear and well-structured, presenting the results in an orderly and careful manner.

2. The manuscript contains excessive mention of previous self-studies.

3. The material and methodology are stated correctly.

4. Tables and figures are easy to interpret and understandably described. 

5. The conclusions need to be rewritten to indicate the main theses of the paper and evidence of the Authors' statements.

Specific comments:

1. The manuscript requires the identification of a specific research problem and research hypothesis. 

2. The discussion should be enriched with the position of the last few years of research centers from around the world.

3. In the conclusions, the authors should focus on the main theses of the paper, rather than stating the conclusions of the previous article. After pointing out the weaknesses/limitations of the research, it would be advisable to indicate specific (not just general) actions to eliminate them.

4. Please justify the choice of statistical test for the sample data.

Author Response

General concept comments:

  1. The manuscript is clear and well-structured, presenting the results in an orderly and careful manner.

Thank you.

  1. The manuscript contains excessive mention of previous self-studies.

We have deleted “The subsequent in-house study found that the level of those lignans was too low to be detected in serum from rats treated with the lignan rich fraction (SWCA) of S. williamsii for 10 weeks, suggesting that lignans might be metabolized by intestinal microbiota before absorption and that they might exert their bone protective effects via activating intestinal targets.” in the Introduction.

  1. The material and methodology are stated correctly.

Thank you.

  1. Tables and figures are easy to interpret and understandably described. 

Thank you.

  1. The conclusions need to be rewritten to indicate the main theses of the paper and evidence of the Authors' statements.

The conclusions have been rewritten accordingly.

Specific comments:

  1. The manuscript requires the identification of a specific research problem and research hypothesis. 

The hypothesis of “Based on our previous studies, we hypothesized that SWCA could modulate 5-HT synthesis via gut TPH-1 and microbial composition to exert protective effects on bone indirectly.” has been added to page 3.

  1. The discussion should be enriched with the position of the last few years of research centers from around the world.

The last researches of “In a recent cohort study of 388,979 individuals, Kang et al. confirmed again that the use of serotonin reuptake inhibitors was associated with an increased risk of bone loss in both men and women in average age of 58 [27]. However, a cross-sectional study of 45 women (mean age: 37.64) suffered from major depressive disorder did not show significant differences in the mean BMD in lumber spine, left hip and right hip compared with healthy controls (mean age: 38.1) [28]. The controversial results might be due to the differences in methodology, the age of subjects and the analysis method. In any case, concerns about bone health have been raised when applying SSRIs.” and “A few recent studies reported the modulating actions of lignans on gut microbial composition. Bader et al. [34] revealed that flaxseed lignan secoisolariciresinol diglucoside altered 3 bacterial phyla and 22 genera upon 10-day treatment in eight-week-old female C57BL/6 mice. Taibi et al. [35] found that lignan-rich oilseeds could strongly influence the microbiota composition of younger and premenopausal females, and led to different enterolignan production in an in vitro fermentation experiment.” have been enriched in the Discussion.

  1. Kang, S.; Han, M.; Park, C.I.; Jung, I.; Kim, E.H.; Boo, Y.J.; Kang, J.I.; Kim, S.J. Use of serotonin reuptake inhibitors and risk of subsequent bone loss in a nationwide population-based cohort study. Rep. 2021, 11, 13461-13461.
  2. Ho, R.C.; Chua, A.N.; Husain, S.F.; Tan, W.; Hao, F.; Vu, G.T.; Tran, B.X.; Nguyen, H.T.; McIntyre, R.S.; Ho, C.S. Premenopausal Singaporean Women Suffering from Major Depressive Disorder Treated with Selective Serotonin Reuptake Inhibitors Had Similar Bone Mineral Density as Compared with Healthy Controls. Diagnostics (Basel) 2022, 12, 96.
  3. Badger, R.; Aho, K.; Serve, K. Short‐term exposure to synthetic flaxseed lignan LGM2605 alters gut microbiota in mice. MicrobiologyOpen (Weinheim) 2021, 10, e1185-n/a.
  4. Corona, G.; Kreimes, A.; Barone, M.; Turroni, S.; Brigidi, P.; Keleszade, E.; Costabile, A. Impact of lignans in oilseed mix on gut microbiome composition and enterolignan production in younger healthy and premenopausal women: an in vitro pilot study. Microbial cell factories 2020, 19, 82-82.
  5. In the conclusions, the authors should focus on the main theses of the paper, rather than stating the conclusions of the previous article. After pointing out the weaknesses/limitations of the research, it would be advisable to indicate specific (not just general) actions to eliminate them.

The conclusion has been modified to “Our present rat study indicated that the lignan-rich fraction from S. williamsii might exert bone protective effects indirectly by suppressing serotonin synthesis via inhibiting intestinal TPH-1 protein and modulating gut microbiota composition”. The further studies for limitations have been specified as “The limitation of the present study is that there is no direct evidence for the modulating effects of lignan altered microbiome on bone metabolism and 5-HT synthesis. Further study of SWCA treated-feces transplantation to antibiotic pre-intervened mice is needed to determine the actions of lignan-induced microbiome on bone. In addition, in vitro studies of major lignans from SWCA on 5-HT synthesis and affinity to TPH-1 are necessary to verify the direct actions of lignans on 5-HT.”

  1. Please justify the choice of statistical test for the sample data.

we justified and revised the statistical analysis on page 6 as follows: The data were expressed as mean ± SEM. The data were checked for the assumptions of normality and equal variance, and then performed by GraphPad PRISM software package 8. To evaluate the actions of lignan-rich fraction to alleviate the affects of estrogen deficiency in ovariectomized rats, one-way analysis of variance (ANOVA) followed by Tukey’s multiple comparisons test was performed with a single pooled variance. The re-sults with p-values of < 0.05 were considered statistically significant. The correlation be-tween serum serotonin and BMD was evaluated by linear regression, and the Pearson R2 was calculated. p < 0.05 was considered to be statistically significant.

The 16S rRNA sequence analysis was performed according to previous publications [24,25]. The diversity and richness of operational taxonomic unites (OTUs) were measured by Sobs index and Shannon index using Student’s t-test. The principal coordinate analysis (PCoA) was calculated by Bray Curtis Distance analysis with ANOSIM test. The analysis of significances between groups was performed by Kruskal-Wallis H test followed with Tukey-kramer post hoc test. The clade of microbial community was analyzed by LEfSe tool with LDA scores > 2 and shown as cladogram. The correlation of bone mineral density and microbial species was performed by Spearman correlation heatmap.

Reviewer 2 Report

Manuscript ID: nutrients-1959289

Title: The lignan-rich fraction from Sambucus williamsii Hance exerts bone protective effects via altering circulating serotonin and gut microbiota in rats

 The work covers very important aspects from the nutritional and medical point of view. The wide scope of research, the method of presenting the results and their discussion constitute the high cognitive value of this manuscript. The wide scope of research, the method of presenting the results and their discussion constitute the high scientific value of this manuscript.

 However, some details should be improved and explained. I would like to make some comments that authors could take into account to improve the overall quality of the manuscript.

 Comments:

Abbreviation: the lignan-rich fraction was abbreviate as CA, but why? SWCA means Sambucus williamsii… but where did the first letters of the rest of the abbreviation come from?

Line 135: The quality of CA was characterized, however it was not explain what exactly mean mentioned in manuscript major components (M1 to M4)?

Line 134: The yield of CA was 0.26% (w/w). It means that 1g of dried ramulus of S. williamsii gave 0.23 g CA. 83.3 mg out off 230 mg of CA was described as “major components”. What was the second major component (146.7 mg) of CA? The material should be precisely described/characterised.

Lines 196-199: Why D4-serotonin and D4-L-Kyn was not used as internal standard. The 5-Methoxytryptamine (line 201) was used as internal standard (IS)instead that. This approach is illogical for me. The best IS is compound with the same structure and the same retention time.

Point 2.12 Not all applied statistical analysis and programs were mentioned there.

Units: The some units should be corrected for instance line 269 – it is written R2, but “2” should have been written as superscript, etc.

Table 1. The results of post test was indicated as stars superscript but it is not clear. Look into first line  (BMD) 245.1***, it means that the difference is statistical significant for pair of mean values or for all? It is clear in Fig. 2 because it was mentioned that “#” indicates difference between “p < 0.05 vs. Sham” In my opinion it is better to introduce letters in rows (Tab. 1) the same letters – the significant difference was not observed.

Fig. 3 It was not explained why result for CAH samples were not introduced in Fig. 3.

Point 3.7 and Fig. 7. LefSe analysis was done. The cladogram diagram was presented. A few sentences should be introduce to explain how to read cladogram, which nodes represent species with no significant difference and which nodes with significant difference. The cut-off point of LDA score should be given because it is not known about parameter which is crucial to categorizing the species at the level of phylum, class, order, family, and genus as significant different.

Author Response

  1. Abbreviation: the lignan-rich fraction was abbreviate as CA, but why? SWCA means Sambucus williamsii… but where did the first letters of the rest of the abbreviation come from?

Our work on S.williamsii has been doing over ten years. In this study, the lignan-rich fraction is prepared by extracting using 60% aqueous ethanol and then enriching using 50% aqueous ethanol from resin column. In our previous animal study [14], we identified a bioactive fraction SWC, namely the combined eluate of 50% and 95% aqueous ethanol, from its concomitant fractions SWA and SWB. SWCA is a purified fraction from SWC. In order to avoid any confusion, we did not mention too much about the history.

  1. Line 135: The quality of CA was characterized, however it was not explain what exactly mean mentioned in manuscript major components (M1 to M4)?

Our previous study has identified the lignan components of CA. The reference has been added and listed as [20] in the main text. The exact names of M1 to M4 were listed in 2.1 Reagent and materials.

[20] Xiao, H.-H.; Lv, J.; Mok, D.; Yao, X.-S.; Wong, M.-S.; Cooper, R. NMR Applications for Botanical Mixtures: The Use of HSQC Data to Determine Lignan Content in Sambucus williamsii. J. Nat. Prod. 2019, 82, 1733-1740.

  1. Line 134: The yield of CA was 0.26% (w/w). It means that 1g of dried ramulus of S. williamsii gave 0.23 g CA. 83.3 mg out off 230 mg of CA was described as “major components”. What was the second major component (146.7 mg) of CA? The material should be precisely described/characterised.

On page 3, line 140, we mentioned that “The contents of the four markers in SWCA were determined to be 41.2, 27.2, 8.6 and 6.3 mg/g, respectively”. The amount of these four markers is in SWCA not in raw herbs. CA is a mixture of lignans and more than thirty lignans were identified from it in our previous studies. However, due to the limitation amount of some of them and the bioactivities, M1 to M4 were selected and purified to be the markers. We have added the explanation in the text on page 3.

  1. Lines 196-199: Why D4-serotonin and D4-L-Kyn was not used as internal standard. The 5-Methoxytryptamine (line 201) was used as internal standard (IS)instead that. This approach is illogical for me. The best IS is compound with the same structure and the same retention time.

In the establishment of quantitation analysis method, it’s ideal to use the same matrix as the expected samples to prepare the QC samples and the calibration standards, and do the method validations, in order to avoid system errors and inaccuracy results. However, 5-HT and Kyn as endogenous substances, have high levels in the “blank” matrix, and they can’t be directly used to set up a methodology. D4-serotonin and D4-L-Kyn have the same physicochemical and analytical properties as 5-HT and Kyn, therefore they were applied as “surrogate analyte” to establish the current analysis method. In this case, another compound 5-Methoxytryptamine was selected as internal standard. We have added the explanation on page 5, line 201, “Surrogate analytes” [5] were introduced to avoid the interference from endogenous 5-HT and Kyn on linearity, precision, accuracy, recovery and matrix effects.”

  1. Point 2.12 Not all applied statistical analysis and programs were mentioned there.

We have added all analysis in 2.12 Statistical analysis. “In 16S rRNA sequence analysis, the diversity and richness of operational taxonomic unites (OTUs) were measured by Sobs index and Shannon index using Student’s t-test. The principal coordinate analysis (PCoA) was calculated by Bray Curtis Distance analysis with ANOSIM test. The analysis of significances between groups was performed by Kruskal-Wallis H test followed with Tukey-kramer post hoc test. The clade of microbial community was analyzed by LEfSe tool with LDA scores > 2 and shown as cladogram. The correlation of bone mineral density and microbial species was performed by Spearman correlation heatmap.”

  1. Units: The some units should be corrected for instance line 269 – it is written R2, but “2” should have been written as superscript, etc.

The units have been carefully checked and revised in the full text.

  1. Table 1. The results of post test was indicated as stars superscript but it is not clear. Look into first line  (BMD) 245.1***, it means that the difference is statistical significant for pair of mean values or for all? It is clear in Fig. 2 because it was mentioned that “#” indicates difference between “p < 0.05 vs. Sham” In my opinion it is better to introduce letters in rows (Tab. 1) the same letters – the significant difference was not observed.

We have revised the significance indicates in Table 1.

  1. 3 It was not explained why result for CAH samples were not introduced in Fig. 3.

We have added the sentence of “As low dose of SWCA exerted more significant effects on bone than the high dose of SWCA, CAL was selected for further signalling pathway study.” on page 8, line 342.

  1. Point 3.7 and Fig. 7. LefSe analysis was done. The cladogram diagram was presented. A few sentences should be introduce to explain how to read cladogram, which nodes represent species with no significant difference and which nodes with significant difference. The cut-off point of LDA score should be given because it is not known about parameter which is crucial to categorizing the species at the level of phylum, class, order, family, and genus as significant different.

The LDA score has been added on page 13 as “In order to identify the clade of Antinobacteria phylum, the microbial community from phylum to genus level was analyzed by LEfSe tool with LDA score >2.” The explain for reading cladogram have been added in the legend of Figure 7 as “cladogram of LEfSe analysis from phylum to genus with LDA score >2, nodes of the same color indicate microbial taxa that is significantly enriched in the corresponding group and has a significant impact on the difference between groups, while the light yellow nodes indicate microbial taxa that do not differ significantly in different groups or have no significant effect on differences between groups;”

Reviewer 3 Report

Manuscript titled "The lignan-rich fraction from Sambucus williamsii Hance exerts bone protective effects via altering circulating serotonin and gut microbiota in rats". The present manuscript is scientifically weak and not technically sound. The results shown especially for evaluation of bone parameters are not enough. The 3d figure (A) view (angle) for OVX and sham are not having same alignment. Standard histology is missing (TRAP staining for osteoclast, massion trichrome for osteoblast) specific bone formation and resorption markers by IF staining not provided.

Author Response

Manuscript titled "The lignan-rich fraction from Sambucus williamsii Hance exerts bone protective effects via altering circulating serotonin and gut microbiota in rats". The present manuscript is scientifically weak and not technically sound. The results shown especially for evaluation of bone parameters are not enough. The 3d figure (A) view (angle) for OVX and sham are not having same alignment. Standard histology is missing (TRAP staining for osteoclast, massion trichrome for osteoblast) specific bone formation and resorption markers by IF staining not provided.

 S. williamsii and its lignan-rich fraction have been demonstrated to be effective against OVX-induced bone loss in our series previous studies (1-4). The present study focus on understanding the mechanism of actions and only employ micro-CT and bone markers to confirm the effects of the fractions on bone but not study the effects using standard histology. We replaced the 3D structure of Sham group in the Figure 1A and added bone resorption marker of CTX-I in Table 1 according to your suggestions.

  1. Xie, F.; Wu, C.F.; Zhang, Y.; Yao, X.S.; Cheung, P.Y.; Chan, A.S.; Wong, M.S. Increase in bone mass and bone strength by Sambucus williamsii Biol. Pharm. Bull. 2005, 28, 1879-1885.
  2. Xiao, H.H.; Dai, Y.; Wan, H.Y.; Wong, M.S.; Yao, X.S. Bone protective effects of bioactive fractions and ingredients in Sambucus williamsii Br. J. Nutr. 2011, 106, 1802-1809.
  3. Xiao, H.H.; Sham, T.T.; Chan, C.-O.; Li, M.H.; Chen, X.; Wu, Q.C.; Mok, D.K.W.; Yao, X.S.; Wong, M.S. A Metabolomics Study on the Bone Protective Effects of a Lignan-Rich Fraction From Sambucus Williamsii Ramulus in Aged Rats. Pharmacol. 2018, 9, 932-932.
  4. Zhang, Y.; Li, Q.; Wan, H.Y.; Xiao, H.H.; Lai, W.P.; Yao, X.S.; Wong, M.S. Study of the mechanisms by which Sambucus williamsii HANCE extract exert protective effects aginst ovariectomy-induced osteoporosis in vivo. Int. 2011, 22, 703-709.

Round 2

Reviewer 2 Report

The paper has been corrected significantly and I think that final version of this paper can be considered for publication.

Reviewer 3 Report

The revised manuscript is scientifically weak and not technically sound. The results shown especially for the evaluation of bone parameters are not satisfactory.